# COVID-19 and Respiratory Virus Co-Infections: A Systematic Review of the Literature

**DOI:** 10.3390/v15040865

**Published:** 2023-03-28

**Authors:** Helena C. Maltezou, Amalia Papanikolopoulou, Sofia Vassiliu, Kalliopi Theodoridou, Georgia Nikolopoulou, Nikolaos V. Sipsas

**Affiliations:** 1Directorate of Research, Studies and Documentation, National Public Health Organization, 15123 Athens, Greece; 2Third Department of Internal Medicine, Sotiria General Hospital for Thoracic Diseases, National and Kapodistrian University of Athens, School of Medicine, Sotiria General Hospital, 11527 Athens, Greece; 3Nafplion General Hospital, 21100 Nafplion, Greece; 4Department of Microbiology, Andreas Sygros Hospital, National and Kapodistrian University of Athens, 16121 Athens, Greece; 5Department of Hepatitides, National Public Health Organization, 15123 Athens, Greece; 6Pathophysiology Department, Medical School, National and Kapodistrian University of Athens, 11527 Athens, Greece

**Keywords:** COVID-19, SARS-CoV-2, influenza, RSV, respiratory virus, co-infection

## Abstract

Τhe COVID-19 pandemic highly impacted the circulation, seasonality, and morbidity burden of several respiratory viruses. We reviewed published cases of SARS-CoV-2 and respiratory virus co-infections as of 12 April 2022. SARS-CoV-2 and influenza co-infections were reported almost exclusively during the first pandemic wave. It is possible that the overall incidence of SARS-CoV-2 co-infections is higher because of the paucity of co-testing for respiratory viruses during the first pandemic waves when mild cases might have been missed. Animal models indicate severe lung pathology and high fatality; nevertheless, the available literature is largely inconclusive regarding the clinical course and prognosis of co-infected patients. Animal models also indicate the importance of considering the sequence timing of each respiratory virus infection; however, there is no such information in reported human cases. Given the differences between 2020 and 2023 in terms of epidemiology and availability of vaccines and specific treatment against COVID-19, it is rational not to extrapolate these early findings to present times. It is expected that the characteristics of SARS-CoV-2 and respiratory virus co-infections will evolve in the upcoming seasons. Multiplex real-time PCR-based assays have been developed in the past two years and should be used to increase diagnostic and infection control capacity, and also for surveillance purposes. Given that COVID-19 and influenza share the same high-risk groups, it is essential that the latter get vaccinated against both viruses. Further studies are needed to elucidate how SARS-CoV-2 and respiratory virus co-infections will be shaped in the upcoming years, in terms of impact and prognosis.

## 1. Introduction

Although the first COVID-19 vaccines became available less than one year after the declaration of the pandemic in March 2020, as of early 2023, COVID-19 remains a leading infectious cause of hospitalization and deaths globally, especially in individuals ≥80 years old, which can be prevented by vaccination [1,2]. The persisting COVID-19-associated morbidity and adverse outcomes are mainly attributed to the waning vaccine-derived immunity, the emergence of highly transmissible SARS-CoV-2 subvariants with immune-escape capacity, and also the fact that several individuals remain unvaccinated for various reasons [3,4,5]. Seasonal influenza (hereafter referred to as influenza) is also a leading cause of morbidity, mortality, and use of healthcare services globally. A meta-regression analysis of 34 randomized control trials extending over 47 pre-COVID-19 influenza seasons (1970 through 2009) found that the annual attack rate of all influenza was 3.5% [95% confidence intervals (CI): 2.30–4.60%] for unvaccinated adults and 15.2% (95% CI: 11.40–18.90%) for unvaccinated children [6]. Similarly, the morbidity burden of acute respiratory infections associated with respiratory syncytial virus (RSV) is substantial and is confined mainly to infants and older adults [7,8].

During the first COVID-19 pandemic wave in early 2020, because of the co-circulation of SARS-CoV-2, influenza, and other respiratory viruses, cases of COVID-19 and respiratory virus co-infections were described [9,10,11,12,13,14]. However, early studies on COVID-19 co-infections were often inconclusive regarding severity and outcome. Nonetheless, a recently published United Kingdom (UK) study based on a database of 212,466 SARS-CoV-2-infected adults hospitalized between 6 February 2020 and 8 December 2021 found that compared to SARS-CoV-2 mono-infected patients, SARS-CoV-2 and influenza co-infected patients had increased odds of receiving invasive mechanical ventilation and increased in-hospital mortality [15].

The next two COVID-19 pandemic seasons (2020–2021 and 2021–2022) were dominated by the emergence of the more transmissible Delta and Omicron SARS-CoV-2 variants with immune escape capacity [4], while influenza had a negligible epidemiological and clinical impact if at all [16,17]. The circulation and morbidity burden of other respiratory viruses [e.g., RSV, human metapneumovirus (hMPV), adenoviruses, rhinoviruses] were also highly impacted by the COVID-19 pandemic, with wide temporal and geographic fluctuations [18,19,20,21,22,23,24,25,26].

After the non-pharmaceutical interventions (NPIs) were lifted almost universally in 2022, annual influenza epidemics returned; however, with notable differences in terms of duration, peak, and morbidity burden compared to pre-COVID-19 seasons [27,28,29]. It is highly possible that COVID-19 and influenza epidemics will occur concurrently in the upcoming years [30,31]. Given that COVID-19 and influenza share the same high-risk groups, co-infection with SARS-CoV-2 and influenza viruses could represent a serious threat to them [32]. In addition, COVID-19 and respiratory virus co-infections may have healthcare and public health implications. These include COVID-19 and influenza vaccination campaigns, preparation of healthcare systems to address high healthcare demand and hospitalizations, diagnostic testing, empiric antiviral treatment, and infection prevention and control practices [32,33]. Moreover, soon, newer vaccines against RSV will be available for the elderly and newborns (via maternal vaccination) to add to the list of influenza and COVID-19 vaccines. Potential logistical issues should be addressed unless combination vaccines become available. This is a systematic review of the current state of knowledge of COVID-19 and respiratory virus co-infections, with an emphasis on epidemiological characteristics and clinical outcomes.

## 2. Methods

This study was conducted as a systematic review in accordance with the Preferred Reporting Items for Systematic Reviews and Meta-Analyses (PRISMA) guidelines. We searched the following databases: PubMed and official websites. Databases were searched using the following search strings that included both MeSH terms and text words. In particular, PubMed was searched for articles published from inception tο 12 April 2022, using the following search strings that included both MeSH terms and text words: COVID-19AND influenza, COVID-19ANDparainfluenza, COVID-19ANDRSV, COVID-19ANDadenovirus, COVID-19ANDhMPV, COVID-19ANDrhinovirus, and COVID-19ANDbocavirus. Studies were selected and screened based on title and abstract using the inclusion/exclusion criteria. After screening, all selected full-text articles were assessed for eligibility. Inclusion criteria included: (a) patients with confirmed COVID-19 and (b) patients with confirmed viral co-infection. Only laboratory-confirmed cases were studied. Articles reporting only bacterial or fungal co-infections were not considered. Studies were restricted to the English language since study resources precluded any translation activities. We included only published data in the systematic review, whereas unpublished data such as abstracts and posters were excluded.

We screened the abstracts of all 5296 articles identified through this first round of searching and selected 83 articles based on our inclusion/exclusion criteria. Information from 45 additional articles and official public health websites was also used. Overall, 128 articles were included in the review (Figure 1).

In order to estimate the mean and median incidence of co-infections (e.g., for influenza co-infection) we used only original studies reporting the denominator of those tested for both viruses.

## 3. SARS-CoV-2 and Influenza Co-Infections

Appendix A summarizes articles reporting cases of SARS-CoV-2 and influenza virus co-infections, published as of 12 April 2022. SARS-CoV-2 and influenza co-infections have been described as case reports, case series, single or multi-center studies, cohorts of community or hospitalized cases, as well as within the frame of sentinel and laboratory surveillance at national laboratories or healthcare facilities [9,10,11,12,13,14,15,25,27,34,35,36,37,38,39,40,41,42,43,44,45,46,47,48,49,50,51,52,53,54,55,56,57,58,59,60,61,62,63,64,65,66,67,68].

Almost all SARS-CoV-2 and influenza virus co-infections were detected during the first COVID-19 pandemic wave or by December 2020, following the seasonal pattern of influenza at that time (Appendix A). In particular, early studies from Wuhan, China found that COVID-19 patients were frequently co-infected with influenza viruses A or B, depending on the predominant influenza type in the specific time period [34,35]. For instance, in mid-January 2020, COVID-19 patients hospitalized in Wuhan were almost always co-infected with influenza B; however, after 28 January 2020, influenza A was the prevalent co-pathogen [34]. Another Chinese study from 28 January through 24 March 2020 found that 45.5% (97 out of 213) of COVID-19 patients tested positive for IgM antibodies against influenza [36]. In a similar line, an Iranian study showed that almost one in every four dead COVID-19 patients also tested positive for influenza (mostly A/H1N1), while other respiratory viruses (e.g., RSV, bocavirus, parainfluenza viruses, hMPV, adenovirus) were also detected in several cases [11]. Given the temporal overlap of SARS-CoV-2 and influenza in the first months of the COVID-19 pandemic and their clinical similarity [10,37], it is possible that several co-infections remained undetected, especially in those with milder courses not attending healthcare services. It is also possible that SARS-CoV-2 transmission was initially unseen during the influenza epidemics [69,70].

Overall, out of 21 published studies with available data, the mean percentage of influenza co-infection in SARS-CoV-2 infected patients was estimated at 16.3%, ranging from 0.04% to 58.3%, while the median percentage of SARS-CoV-2 and influenza co-infections was 4.9% [10,11,15,27,34,35,36,38,40,42,49,54,55,57,59,61,63,64,66,67,68]. The incidence of influenza co-infections diminished as the first COVID-19 pandemic wave evolved. For instance, a retrospective study at the Johns Hopkins diagnostic laboratory of nasopharyngeal respiratory specimens collected from inpatients and outpatients between December 2019 and March 2020 found an overall positivity rate of 10.1% for influenza A and 9.9% for influenza B [18]. After March 2020, SARS-CoV-2 predominated, while influenza virus co-infections were extremely uncommon [18]. Overall, during 2020–2021 only 0.18% of respiratory specimens tested positive for influenza in United States (US) clinical laboratories, compared to 10–19% in previous years [16,17].

Patients with SARS-CoV-2 and influenza co-infections had a mean age of 52.5 years (range: two weeks to 81 years) [9,10,12,13,14,36,37,38,39,40,41,42,43,44,45,46,47,48,50,51,52,53,56,57,58,61,62,63,64]. A retrospective study of bacterial co-infections in 1243 SARS-CoV-2-infected, 775 influenza-infected, or 242 RSV-infected adult patients with community-acquired pneumonia admitted to Karolinska University Hospital in Stockholm, Sweden, from 2011 through 2020, found that SARS-CoV-2 patients were younger than the other two groups (median ages: 62, 69, 71 years, respectively) [71]; however, these findings may be attributed to the prevalent profile of COVID-19 patients in the first pandemic wave [72]. Overall, there are scarce data on age differences between SARS-CoV-2 only infected and SARS-CoV-2 and influenza co-infected patients. A Chinese study of COVID-19 and influenza co-infections reported that COVID-19 patients had a median age of 47 years compared with a median of 50 years among influenza A/HINI-infected patients [63]. In contrast, a Korean study of 436 COVID-19 patients (35 patients were co-infected with influenza virus while 401 patients had SARS-CoV-2 only) found that co-infected patients were significantly older than SARS-CoV-2 only infected patients (median age 70 versus 46 years of age, *p*-value < 0.001) [67]. Overall, males were slightly overrepresented among SARS-CoV-2 and influenza co-infected published cases, accounting for a mean of 52.9% [9,10,12,13,14,36,37,38,39,40,41,42,43,44,45,46,47,48,50,51,52,53,56,57,61,62,63,64].

Review of available evidence indicates that co-morbidities were present in 59.89% of SARS-CoV-2 and influenza co-infected patients [9,10,11,12,13,14,15,27,34,35,36,37,38,39,40,41,42,43,44,45,46,47,48,49,50,51,52,53,54,55,56,57,58,59,60,61,62,63,64,65,66,67]. The most prevalent co-morbidities among co-infected patients were obesity, diabetes mellitus, hypertension, malignancy, and chronic cardiovascular disease [9,10,13,14,38,39,40,42,43,44,45,46,47,50,51,52,57,58,61,62].

Overall, 82.3% of SARS-CoV-2 and influenza co-infected patients were symptomatic at diagnosis [9,10,11,12,13,14,15,27,34,35,36,37,38,39,40,41,42,43,44,45,46,47,48,49,50,51,52,53,54,55,56,57,58,60,61,62,63,64,65,66,67]. Despite the differences in terms of transmissibility and incubation period between SARS-CoV-2 and influenza viruses, co-infected patients usually manifested symptoms typical of COVID-19, including (new or persisting) fever (93.2%), cough (97.6%), (new or aggravated) shortness of breath/dyspnea (95.0%), diarrhea (93.8%), myalgia (44.1%), and fatigue (40.3%) [9,10,12,13,14,34,35,36,37,38,39,40,41,42,43,44,45,46,47,48,50,51,52,53,58,61,62,63]. Differences in incidence rates of specific symptoms have been occasionally recorded, e.g., early Chinese studies found that SARS-CoV-2 and influenza co-infected patients were more likely to develop fever, and/or myalgia than COVID-19 patients [35,63]. In one of these studies, critically ill patients with COVID and influenza co-infection were more prone to cardiac injury than those without influenza (86.4% versus 54.5%, *p*-value = 0.04), but not to other complications [35]. Another study showed that SARS-CoV-2 and influenza co-infected patients were significantly more likely to develop acute kidney injury and acute heart failure compared with COVID-19 patients [63]. Overall, cardiac injury was reported in 60.2% of 83 SARS-CoV-2 and influenza virus co-infected cases [14,35,63]. Nevertheless, the most overt complication of SARS-CoV-2 and influenza co-infected cases was acute hypoxemic respiratory failure and respiratory distress syndrome [9,10,11,12,13,14,34,35,36,37,38,39,40,41,42,43,44,45,46,47,48,49,50,51,52,53,56,58,61,62].

A Korean study of 436 adult COVID-19 patients, of whom 35 had influenza co-infection and 401 were infected with SARS-CoV-2 only, found that co-infected patients were hospitalized more often (85.7% versus 6.7%; *p*-value < 0.001) and had significantly higher inflammatory markers (C-reactive protein [CRP], procalcitonin, and lactate dehydrogenase (LDH); *p*-values = 0.019, <0.001, and 0.024) but significantly lower lymphocyte counts [*p*-value = 0.003] [67]. However, a systemic review and meta-analysis of 12 studies up to July 9, 2021 (8 from China, 1 from Iran, 1 from USA, 1 from England, and 1 from Saudi Arabia) with a total of 9498 patients found that lymphocyte counts were significantly higher in SARS-CoV-2 and influenza co-infected patients, while activated partial thromboplastin time (APTT) was significantly more prolonged in co-infected patients than in SARS-CoV-2 infected patients (*p*-values = 0.04 and 0.007, respectively); however, most studies found no laboratory differences between SARS-CoV-2 infected and influenza co-infected patients in terms of white blood cell (WBC) counts, CRP, and interleukin-6 (IL-6) [73].

Chest radiographic findings of COVID-19 and influenza co-infected patients showed infiltrates, mild vascular congestion, and patchy diffuse bilateral infiltrates, while chest computed tomography (CT) findings were mostly compatible with COVID-19, demonstrating bilateral multifocal ground-glass opacities with peripheral distribution and mild interlobular septal thickening [9,10,11,12,13,14,34,35,36,37,38,39,40,41,42,43,44,45,46,47,48,49,50,51,52]. However, an early Chinese study found that SARS-CoV-2 only infected patients younger than 60 years had higher CT scores than patients with influenza A co-infection [36]. In the latter study, increased expression of serum cytokines IL-2, IL-6, and IL-8, tumor necrosis factor-α (TNF-α), and cardiac troponin I was documented in patients with SARS-CoV-2 infection only, which indicates that in this case series, influenza A had no serious impact on disease outcomes, but rather reduced inflammation in certain COVID-19 patients [36]. In contrast, another Chinese study found that compared with COVID-19 patients, COVID-19 and influenza co-infected patients had more profound inflammation and organ injury, as expressed by significantly higher WBCs, neutrophil counts, creatinine levels, D-dimers, and TNF-α (*p*-values < 0.05), indicative of a “cytokine storm” triggered by influenza co-infection [35]. Another study of 131 COVID-19 patients and 176 COVID-19 and influenza A or B co-infected patients hospitalized from 21 January through 21 February 2020 in Wuhan, China found no difference between the three groups in terms of age, sex, severity of illness at admission, and laboratory findings [34]. In this latter study, most patients had fever and cough, regardless of influenza virus infection; however, SARS-CoV-2 co-infected patients with influenza B had more frequent fatigue (13%), chest CT abnormalities (100%), or decreased lymphocyte and eosinophil counts, indicating a more severe clinical course [34]. In contrast, SARS-CoV-2 and influenza A co-infected patients had a milder course, higher lymphocyte and eosinophil counts, and fewer abnormalities on chest CT [34]. Although all patients were treated similarly, those with SARS-CoV-2 and influenza B co-infection were more likely to have a poorer prognosis (defined as aggravated disease or death) compared with influenza A co-infected patients or COVID-19 patients only (30.4%, 5.9%, and 7.6%, respectively) [34]. Another study found that SARS-CoV-2 and influenza co-infected patients were significantly more likely to develop multilobar infiltrates and secondary bacterial infections, have a severe course, be admitted to an intensive care unit (ICU) (13.9% versus 5.2%; *p*-value = 0.046), and die compared with COVID-19 patients (2.8% versus 0% mortality, *p*-value = 0.008) [63].

Published evidence indicates a mean rate of admission to an ICU of 6.2% among SARS-CoV-2 and influenza co-infected patients [9,13,14,34,37,39,40,41,42,44,45,46,47,48,50,51,53,56,57,58,60,62,63]. SARS-CoV-2 and influenza co-infections have been associated with increased odds of acute kidney injury, acute heart failure, secondary bacterial infections, multilobar infiltrates, and admission to ICU compared with SARS-CoV-2 only infected patients [35,63]. Moreover, the mean case fatality rate of SARS-CoV-2 and influenza co-infected patients was estimated at 12.3% [9,11,14,34,35,36,39,40,41,42,45,46,47,48,49,50,51,52,53,56,57,58,60,61,62,63,64,65,66,67,68,69,70,71,72,74]. Early series from Wuhan, China indicated no difference in outcomes among influenza co-infected COVID-19 patients and those with COVID-19 only [35]. Similarly, others have shown that SARS-CoV-2 co-infections [bacterial or viral] were not an independent factor of severity on admission, need for invasive mechanical ventilation, and mortality [25,59,64,65]. However, another study found that older age, co-infection, and elevated LDH were associated with increased risk for mortality (odds ratios [ORs]: 6.095, 1.089, and 1.006, respectively) [67]. A UK study of 6965 hospitalized COVID-19 patients of whom 583 (8.4%) had laboratory evidence of a respiratory virus co-infection (227 with influenza, 220 with RSV, and 136 with adenovirus) also demonstrated that influenza co-infected patients were more likely to receive invasive mechanical ventilation (OR: 4.14, 95% CI: 2–8.49, *p*-value = 0.0001) or to die (2.35, CI: 1.07–5.12, *p*-value = 0.031) compared with patients with SARS-CoV-2 only [15]. Similarly, influenza A was the only pathogen for which a direct association with mortality was found among 48 hospitalized COVID-19 patients, of whom, 17 (35.5%) had an influenza co-infection [54]. Lastly, a systemic review and meta-analysis of SARS-CoV-2 and influenza virus co-infections among 9498 COVID-19 patients up to 9 July 2021 found no association between co-infection and mortality, although mortality significantly varied by geographic region [73]. In this review, an overall lower risk for critical outcomes was estimated among co-infected patients than SARS-CoV-2 only infected patients (OR=0.64, CI: 0.43–0.97) [73]. Regarding children, two studies have shown that SARS-CoV-2 co-infected children more frequently required oxygen support than those with COVID-19 only, but there was no difference between those with co-infection and those with COVID-19 only in terms of ICU admission and mortality [25,74].

## 4. SARS-CoV-2 and Respiratory Virus Co-Infections Other Than Influenza

Appendix A summarizes articles reporting cases of SARS-CoV-2 and respiratory virus co-infections other than influenza published as of 12 April 2022 [18,22,23,24,54,59,66,67,75,76,77,78,79,80,81,82,83,84,85,86,87,88,89,90,91,92,93,94,95,96,97,98,99,100,101,102,103]. All papers concern the first COVID-19 pandemic wave, drawing data from no later than April 2020. The associated incidence varies greatly among articles. Several studies found that COVID-19 and respiratory virus co-infections other than influenza are common [59,80,81] given their concomitant circulation in the community [82]. For instance, one Saudi Arabia study screening 48 COVID-19 hospitalized patients for 24 respiratory pathogens through six multiplex PCR panels found evidence of co-infection in 34 (71%) patients [54]. Beyond influenza A/H1N1 and Chlamydia pneumoniae (17 and 13 cases, respectively), adenovirus was detected in ten cases [55]. Zhu et al. reported viral, bacterial, and fungal co-infections in 94.2% of 257 COVID-19 patients (91.8% bacterial: mostly *Streptococcus pneumoniae*, *Klebsiella pneumoniae*, and *Haemophilus influenzae*; 31.5% viral; 23.3% fungal) [104]. One systematic review estimated a pooled prevalence of respiratory co-pathogens of 11.6% (and 26.8% in studies using serum antibody tests), approximately equally distributed between viral and atypical bacteria, with the most common pathogens being influenza viruses and *Mycoplasma pneumoniae* [104]. In hospitalized COVID-19 patients, co-infections comprised 7% bacterial (commonest: *M. pneumoniae*, *Pseudomonas aeruginosa*, *H. influenzae*)—with a significantly higher proportion in the ICU compared to mixed hospital setting—and 3% viral (commonest: RSV and influenza A) [105]. A significant proportion (13.9%) of patients with COVID-19 and hypoxemic pneumonia was also found on hospital admission to be co-infected with influenza H1N1, human coronavirus 229E, rhinovirus, and methicillin-susceptible *Staphylococcus aureus* [83]. A similar percentage of COVID-19 pneumonia cases were found positive for other respiratory pathogens (mainly *M. pneumoniae* and RSV) [84]. Pediatric studies indicate even higher co-infection rates. For instance, in one study, co-infection of children with COVID-19 reached 33%; in this study, *M. pneumoniae* accounted for almost 75% of detected pathogens, followed by viruses (influenza, RSV, and adenovirus among others) [85]. A 2021 US multi-center study found that 15.8% of 713 hospitalized children with COVID-19 had a respiratory virus co-infection, which was even higher in the younger age groups (32.4% of <1-year-old children, 36.1% of 1 to 4-year-old children, 4.2% of 5 to 11-year-old children, and 2.2% of 12 to 17-year-old children; *p*-value <0.001) [74]. RSV accounted for two of every three co-infected children (73% among co-infected children <1-year-old) [74].

Other studies found that SARS-CoV-2 and respiratory virus co-infections other than influenza are uncommon or very rare [18,22,23,67,76,77,78]. According to Nowak et al., co-infections with other respiratory viruses accounted for only 3% of SARS-CoV-2 infected patients (mostly other *coronaviridae*) [86]. Another large study of 3757 SARS-CoV-2-positive individuals revealed a prevalence of 1.5% of respiratory virus co-infections (mostly rhinovirus and enteroviruses); however, in the latter study, there was no concurrent testing for other pathogens in an eight-month period [87]. A Canadian study of 4818 SARS-CoV-2-positive respiratory specimens (94.8% from adults) found evidence of virus co-infection in 134 specimens (2.8%), while children were more likely to develop virus co-infections than adults (10% versus 2.4%; *p*-value < 0.01) [22]. In this study, enterovirus, which was mostly detected in children, was the only virus that retained its seasonal pattern compared with pre-pandemic seasons [22]. Similarly, an Italian study testing, through multiplex real-time PCR, more than twelve thousand hospitalized adults with respiratory symptoms from January 2017 through May 2021, revealed a significant reduction in the overall positivity rate during the study period (14.6% in the pre-pandemic period versus 2.7% in the COVID-19 pandemic; *p*-value < 0.0001) [23]. In particular, positivity for influenza A and B, hMPV, parainfluenza virus, RSV, and human coronaviruses decreased significantly during the pandemic; in contrast, a significant increase was recorded for rhinovirus only (3.8% versus 5.6%; *p*-value = 0.02) [23]. Of note, co-infections with different respiratory viruses were observed in the pre-pandemic period; however, during the COVID-19 pandemic, the only co-infection detected was SARS-CoV-2 and rhinovirus [23]. Similarly, in a pediatric Italian study among children <2 years old with lower respiratory tract infections revealed elimination of RSV, hMPV, influenza, and parainfluenza infections during the COVID-19 pandemic, but no change in rhinovirus or other human coronaviruses activities [24].

Differences in the prevalence of respiratory virus co-infections other than influenza in COVID-19 patients are attributed to the particular setting and the season of each study [18]. The endemic profile of different pathogens may vary between hospital versus community cases, by geographic region and age group, and may also fluctuate throughout the year. Moreover, after the onset of the COVID-19 pandemic in late 2019, the implementation of NPIs constantly—which continuously fluctuate worldwide—altered the trajectory of all infections (especially the respiratory ones). A Southern China study of circulation patterns of 11 respiratory viruses (influenza A, influenza B, human MPV, parainfluenza virus, adenovirus, rhinovirus, RSV, and four human coronaviruses) through routine testing of 58,169 throat swabs from 1 January 2018 to 31 December 2020 revealed that the overall activity of respiratory viruses was lower during the implementation of the stringent NPIs in 2020; however, virus activity rebounded shortly after the relaxation of the NPIs and the return of social activities [19]. Nonetheless, this was not the case for rhinovirus, for which the activity remained even during the period of stringent NPIs [19]. A 2018–2020 Chinese study of children aged 28 days to 15 years old revealed a significantly lower rate of detection in 2020 compared with 2018–2019 of hMPV and RSV (*p*-values < 0.001 and 0.01, respectively), but an increase in rhinovirus detection (*p*-value < 0.0010) [20]. RSV was the most prevalent detected respiratory virus in 2018 and 2019 (accounting for 8.6% and 6.7% of respiratory infections), but in 2020, human rhinovirus prevailed (8.7% of respiratory infections) [20]. Co-infections were also reduced during the COVID-19 pandemic (8.6% out of 1948 infections in 2018, 11.6% out of 1796 infections in 2019, and 3.6% out of 1559 in 2020, *p*-value < 0.001). Overall, respiratory infections were limited in both infants and children older than five years during 2020 compared with 2018–2019; however, no change in terms of age, sex, duration of hospitalization, incidence of fever, dyspnea, and wheezing was noted among children with co-infections in 2018, 2019, or 2020 [20]. Similarly, a US retrospective study of 745 valid SARS-CoV-2-positive samples, disclosed that 53 (7.1%) samples were also positive for one or more respiratory viruses, mostly rhinovirus/enterovirus (22 samples; 41.5%), hMPV (18 samples; 33.9%), and adenovirus (12 samples; 22.6%), while influenza A or B was not detected [103]. The median age of respiratory virus co-infected patients was 38 years, while respiratory virus co-infections other than influenza were more common among children up to nine years old [103]. In the pediatric age group, hMPV, RSV, bocavirus, and rhinovirus/enterovirus predominated, while RSV, bocavirus, and rhinovirus/enterovirus were only detected in pediatric samples [103].

Regarding the demographic characteristics of patients with respiratory virus co-infections other than influenza, two studies indicated a mean age of 60 years old [86,87]. However, other studies indicated younger ages. For example, Si et al. found the mean age to be approximately 45 years old [89]. Another large study found that co-infection rates were highest in ages 15–44 years and lowest in ages below 15 years [81]. Moreover, a large study of nasopharyngeal swab specimens tested for a large panel of respiratory pathogens (adenovirus, hMPV, enterovirus/rhinovirus, influenza A, influenza B, parainfluenza 1–4, RSV A and B, *C. pneumoniae*, *M. pneumoniae*, HCoV-229E, HCoV-HKU1, HCoV-NL63, HCoV-OC43) from five hospitals and 40 ambulatory care offices from May through October 2021, and revealed that 63.5% of patients with SARS-CoV-2 and virus co-infections were up to two-years-old [18]. No significant difference between males and females was noted in the literature [80,81], except for one study which showed a male predilection [87].

In terms of clinical findings and patients’ outcomes, no significant difference was observed between the SARS-CoV-2 only infected and co-infected groups in most studies [70,80,81]. According to Zhu et al., most co-infections occurred within 1–4 days of the onset of COVID-19 disease [81]. Most reported symptoms in the co-infection group were fever and cough, followed by other influenza-like symptoms: fatigue, sore throat, rhinorrhea, diaphoresis, arthralgia, shortness of breath, and decreased oxygen O_2_ saturation [85,88,90,91,92,93,94,95,96]. In children, diarrhea, vomiting, and decreased appetite were also recorded [102,103,104]. Certain studies also included asymptomatic patients (8.5% of 257 co-infected patients in Zhu et al.) [81].

In terms of laboratory findings, one study found higher levels of procalcitonin in virus co-infected patients with no difference by sex, age, and disease severity [84]. Chest CT findings demonstrated no statistical difference between the co-infected and SARS-CoV-2 mono-infected patients [84] and the most frequently reported CT finding was ground-glass opacities [90,92,100,101]. Likewise, the length of hospitalization, number of ICU admissions, and deaths did not differ significantly between the two groups [80,83,87]. A study found that co-infections in children have no significant impact on the clinical course [85]. However, a US study in March–April 2020 showed that of 306 COVID-19 patients with additional negative tests for influenza and RSV, 14 (4.6%) were positive for a non-influenza virus (mostly human coronavirus, rhinovirus, and parainfluenza 3) [75]. In this study, those with SARS-CoV-2 infection only were more likely to be admitted (73.6% versus 42.9%, *p*-value = 0.01) and to have a severe outcome (defined as invasive mechanical ventilation or death within 30 days) (36.3% versus 21.4% in the co-infected group; *p*-value = 0.24) compared with virus co-infected patients [75]. Other studies using multivariate analyses have shown that co-infections are not an independent factor of clinical severity, need for a high-flow nasal cannula or invasive mechanical ventilation, and death [59]. Lastly, a multicenter study of 713 hospitalized children with COVID-19 (67.5% of whom had at least one co-morbidity) found that co-infected children more frequently required oxygen support than those with COVID-19 only (69% versus 51.2%; *p*-value < 0.001) [74]. Nevertheless, there was no difference between children with virus co-infection and those with COVID-19 only in terms of ICU admission, duration of hospitalization, requirement of extracorporeal membrane oxygenation, and outcome [74].

An Iranian study among 105 dead SARS-CoV-2 infected patients (7.3% case fatality), also detected RSV and bocavirus in 9.7% of patients, parainfluenza viruses in 3.9% of patients, hMPV in 2.9% of patients, and adenovirus in 1.9% of patients [11]. Interestingly, the three hMPV-positive co-infected deceased patients were children (one 13-month-old and two 6-year-olds, two of whom had a history of asthma); their nasopharyngeal swabs were negative for other viruses (bocavirus, adenovirus, RSV, parainfluenza, influenza A and B) [90]. HMPV was also detected by PCR in a 25-year-old male with no underlying disease, who was admitted to the ICU with acute respiratory failure due to COVID-19; fortunately, he recovered uneventfully [102].

## 5. Animal Models of SARS-CoV-2 and Influenza Co-Infection

To provide insight into the underlying pathogenetic mechanisms of SARS-CoV-2 and influenza co-infections, culture cell models and in vivo influenza virus and SARS-CoV-2 co-infected animal models have been developed. Such models have shown that SARS-CoV-2 and influenza co-infection is associated with more severe pathologic lung findings (e.g., increased infiltration of immune cells and alveolar necrosis) and cytokine levels in the lungs compared to single infection, which indicates high susceptibility to severe lung disease and death after SARS-CoV-2 infection [106,107,108,109]. More specifically, all SARS-CoV-2 and influenza virus co-infected mice died [100% mortality] compared to up to 25% and up to 38% among influenza-infected and SARS-CoV-2 only infected mice [108,109]. Lower serum SARS-CoV-2 neutralizing antibody titers and a more prolonged SARS-CoV-2 shedding were demonstrated in co-infected hamsters compared to SARS-CoV-2 mono-infected hamsters [107]. More severe lung disease was demonstrated regardless of concomitant or sequential influenza and SARS-CoV-2 co-infection [107]. These pathologic findings were not present following infection with other respiratory viruses (e.g., RSV, human parainfluenza virus, rhinovirus) [106]. Notably, co-infected mice and ferrets had increased lung influenza virus loads and decreased SARS-CoV-2 loads, which indicates competition between the influenza virus and SARS-CoV-2 [107,108,109,110]. Nonetheless, prior influenza vaccination significantly decreased influenza titers and virus shedding and contributed to a milder course [110]. The importance of influenza vaccination in conferring protection not only against severe influenza but also against severe SARS-CoV-2 infection has been demonstrated in ferrets vaccinated with live attenuated influenza vaccine three days before SARS-CoV-2 infection, which resulted in a significant reduction of SARS-CoV-2 replication and shedding [111].

In vitro models of SARS-CoV-2 and rhinovirus single or co-infections in nasal epithelia found that replication of the former virus was inhibited by primary but not secondary rhinovirus infection, which was modulated by interferon (IFN) induction [112]. It has also been shown that during concomitant infection of nasal epithelium, SARS-CoV-2 interferes with RSV replication kinetics; however, SARS-CoV-2 replication is not influenced by RSV [113]. These findings indicate the importance of considering the sequence of SARS-CoV-2 and respiratory virus co-infections.

## 6. Implications for Upcoming Seasons

Our systematic review indicates that SARS-CoV-2 and respiratory virus co-infections occurred during the first COVID-19 pandemic wave; however, with wide differences in terms of incidence and characteristics. It is possible that the overall incidence of respiratory virus co-infections is higher because of a paucity of co-testing for other pathogens in COVID-19 cases amid the early pandemic waves. On the other hand, the COVID-19 pandemic highly influenced the epidemiology of other respiratory viruses in terms of seasonality, circulation, and morbidity magnitude. Moreover, although on several occasions SARS-CoV-2 and respiratory viruses co-infections did not significantly influence the spectrum of symptoms, the clinical course, and the outcome, the available literature, mostly from early 2020, is highly inconclusive, while animal models indicate a severe lung pathology and outcome. It should be kept in mind, however, that in several cases, co-infections were asymptomatic, which raises the question of whether these co-infections were truly infections causing clinical illness or merely virus detection without clinical consequence. Data about the cycle thresholds (Cts) might provide an insight into this fact. The high prevalence of co-morbidities among SARS-CoV-2 and influenza co-infected patients may be attributed to the fact that most publications concerned more serious and therefore hospitalized cases, while mild cases might have not attended healthcare facilities and therefore missed diagnosis. Similarly, the high prevalence of co-morbidities among co-infected patients may partially explain their increased rate of complications. The receptor for SARS-CoV-2 is the cell surface angiotensin-converting enzyme 2 (ACE2), which is found in many organs (e.g., lungs, kidneys, vascular endothelium) and explains its multi-system impairment [114]. Influenza can also trigger severe inflammation with striking similarities to the SARS-CoV-2-induced inflammation responses leading to multi-organ impairment and complications [115]. The increased rates of kidney, lung, and heart complications in SARS-CoV-2 and influenza co-infected patients compared to SARS-CoV-2 only infected patients may be attributed to the fact that influenza H1N1 also targets ACE2 [115]. Nevertheless, it is also possible that viral interference may confer some kind of cross-protection due to virus competition at the level of IFN induction and inhibition of virus replication [116].

The more complex pathophysiology of SARS-CoV-2 and respiratory virus co-infections and the diagnostic and therapeutic dilemmas underlined the need for recommendations for testing all influenza-like illness cases for SARS-CoV-2, influenza, and RSV [69,117]. Several multiplex real-time PCR-based assays to detect SARS-CoV-2 and several other respiratory viral and bacterial pathogens have been developed over the past two years, demonstrating excellent performance [55,118,119,120,121,122,123,124,125,126]. Multiplex real-time PCR should be used in everyday practice to increase the diagnostic capacity for respiratory pathogens and to contribute to infection control, and also for surveillance purposes [127].

It is expected that the characteristics and prognosis of SARS-CoV-2 and respiratory virus co-infections will evolve during the upcoming seasons. In the meantime, appropriate prevention and management strategies are needed to reduce the associated morbidity burden. However, given the differences between 2020 and 2022–2023 in terms of epidemiological landscape, implementation of NPIs, and the availability of COVID-19 vaccines and specific treatment against COVID-19, it appears rational not to extrapolate these early findings to present times. The overall impact of respiratory virus co-infections in the present and upcoming seasons remains highly unknown in terms of morbidity burden, clinical severity, and outcome. Given that COVID-19 and influenza share the same high-risk groups, it is essential that the latter get vaccinated against both viruses [32]. It is also recommended that masks (especially N95/P2 masks if possible) are used during periods of increased activity of respiratory viruses by those who are vulnerable (elderly/immunocompromised) and especially by all medical and nurse staff and frontline healthcare staff. It would be of public health importance for future studies to elucidate how respiratory virus co-infections will be shaped during the upcoming years.

## 7. Limitations

The present systematic review has the following limitations that should be considered. First, the overwhelming body of knowledge regarding SARS-CoV-2 and respiratory viruses’ co-infections originates from studies conducted before COVID-19 vaccines and specific treatments became available. For instance, early studies indicate that COVID-19 patients were more likely than influenza patients to be admitted to an ICU (31% versus 12%; *p*-value = 0.002) or die in hospital (20% versus 5%; *p*-value = 0.002) [128]. Second, NPIs have been highly lifted over the past one and a half years; therefore, our knowledge will most probably expand in the upcoming seasons. Third, there was a wide spectrum of reviewed studies in terms of population, seasonality, duration, and tested pathogens, while in almost all of them, there was no standard definition for respiratory virus co-infection, e.g., cases tested positive for two viruses within 7 days [60]. Fourth, evidence from animal models indicates the importance of considering the sequence timing of each virus co-infection (preceding, concomitant, or subsequent infection), which might influence Ct values [112]; nevertheless, there is a paucity of such data in reported human cases. We reviewed the literature up until 12 April 2022, when the Omicron wave was still in its early stages. This study was not able to detect any particularities concerning co-infections of patients infected by the Omicron variants and sub-variants. Lastly, almost all cases of SARS-CoV-2 and respiratory virus co-infections were hospitalized, while the morbidity burden and characteristics of co-infections managed in the community remains highly unknown.

## 8. Conclusions

We reviewed the published cases of SARS-CoV-2 and respiratory virus co-infections as of 12 April 2022 using the PRISMA guidelines. Evidence from the first COVID-19 pandemic wave in early 2020 indicates that SARS-CoV-2 and respiratory virus co-infections occurred, often with a wide range of incidence and an inconclusive impact on clinical severity and prognosis. Animal models indicate that SARS-CoV-2 and influenza co-infection is associated with a more complex pathophysiology and severe lung damage than each viral infection alone. It is essential to expand our knowledge about SARS-CoV-2 co-infections in the next years as more information will be available in the future. The use of multiplex PCR assays should become the standard practice for prompt diagnosis as well as for surveillance purposes. Masks should be routinely used during periods of increased activity of respiratory viruses by vulnerable individuals and frontline healthcare personnel.

## Figures and Tables

**Figure 1 viruses-15-00865-f001:**
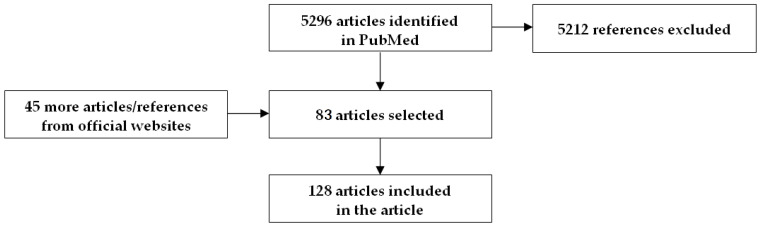
Flow diagram results of literature search.

## Data Availability

Does not apply.

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
