# Peer review of "COVID-19 and Respiratory Virus Co-Infections: A Systematic Review of the Literature"

_viruses, 2023, doi:10.3390/v15040865_

Round 1

Reviewer 1 Report

The authors have tackled the interesting question of co-infections with COVID-19 and other respiratory pathogens with a special focus on influenza co-infections.

A major problem/limitation with this study is that it only goes up to 12 April 2022, a time when many countries had only just seen the return of influenza and other respiratory diseases and so as the authors describe virtually all of the studies selected were during the first year or so of COVID-19 circulation, which limits the relevance of the findings to the current situation where most of the world has various Omicron variants and most of the usual winter/wet season respiratory viruses have returned to circulation albeit some are occurring at irregular times but as Omicron has not settled into a strict winter circulation pattern, these future studies will be more informative than the current study.

Specific comments on the paper.

1. The English standard is not high and the manuscript would be improved by editing by someone with English as their first language. Some word choices for example Abstract first line replace "largely" with "highly or severely"; Abstract mid paragraph, replace "nowadays" with "present times".

2. An example of 1. is the continual misspelling of "systematic" throughout the manuscript.  The authors have peppered the manuscript with the incorrect term "systemic" eg last paragraph of the introduction but it occurs in a number of other places.

3. Either in the introduction or the discussion there needs to be some discussion about the idea of "detection" versus "infection".  As a large proportion of cases of dual "infection" were asymptomatic this raises the question of whether these were truely infections causing a level of disease or merely detection of the pathogen without clinical consequence.  A study of reports which give details of the relative Ct (cycle thresholds) might give some insight into this fact.  For example if the Ct's were low (ie high viral load) with COVID-19 and high Ct's for influenza (ie low viral load) then one might attribute the clinical symptoms to be mainly due to COVID-19 at the time of infection. Of course the timelines for both "infections" might influence these Ct values.

4.  Introduction 3rd line. Add "s" to "death" and this comment should be qualified as most hospitalizations and deaths are in the elderly especially in the over 80 year olds. Introduction last line on the first page, influenza "can effect up to 20% of the population".....   However it rarely does and is usually much lower in the 2-5% range!!

5. Towards the end of the introduction the authors talk about vaccination programs.  It might also be worth mentioning that in the near future vaccines for the elderly and newborns (via maternal vaccination) will be available to add to the list of influenza and COVID-19 vaccines with this added protection but also potentially added logistical issues with the roll out of another respiratory virus vaccine unless combination vaccines are available.

6. The methods the authors have used are a major issue.  They have included many reports with just a single case or just a few cases.  I think these should be deleted from the study as they add very little to the overall conclusions and may sway some figures eg the figure on page 15 which states that the mean percentage of co-infected patients with COVID-19 and influenza was "as low as 0.04% to 100%" a non-sensical conclusion! Eliminating these very low cases will have the added benefit of reducing the massive tables. One suggestion might be to only include studies with say >10 co-infections unless the total subjects analysed were say 50 or 100 subjects with COVID-19.

7. The data in the tables 1 and 2 should also be minimised in the main paper and the full data included in supplementary tables.

8. In the discussion/conclusions apart from recommending vaccinations for COVID-19 and influenza and more multiplex testing, I think the authors should also be recommending the increased use of masks (especially N95/P2 masks if possible) during periods of increased activity of respiratory viruses or during influenza/RSV/COVID-19 "seasons" by those who are vulnerable (elderly/immunocompromised especially) and especially by all medical staff and frontline hospital staff. 

Author Response

Answer to Reviewer 1

Comments and Suggestions for Authors

The authors have tackled the interesting question of co-infections with COVID-19 and other respiratory pathogens with a special focus on influenza co-infections.

A major problem/limitation with this study is that it only goes up to 12 April 2022, a time when many countries had only just seen the return of influenza and other respiratory diseases and so as the authors describe virtually all of the studies selected were during the first year or so of COVID-19 circulation, which limits the relevance of the findings to the current situation where most of the world has various Omicron variants and most of the usual winter/wet season respiratory viruses have returned to circulation albeit some are occurring at irregular times but as Omicron has not settled into a strict winter circulation pattern, these future studies will be more informative than the current study.

Answer to this comment

We thank the reviewer for the valuable comments in order to improve our review. Indeed, we acknowledged in the limitations paragraph, that “the overwhelming body of knowledge regarding SARS-CoV-2 and respiratory viruses’ co-infections originates from studies conducted before COVID-19 vaccines and specific treatment became available.” We also acknowledge that “NPIs have been largely lifted the past one and half year; therefore, our knowledge will most probably expand in the upcoming seasons”. Also “We reviewed the literature up until April 12, 2022, when the Omicron wave was still in the beginning. This study was not able to detect any particularities concerning co-infections of patients infected by the Omicron variants and sub-variants”.

The abovementioned limitations have been discussed extensively. To add the experience of the 2022-2023 season regarding SARS-CoV-2 and respiratory virus co-infections and include studies with Omicron variants, we should expand our review period and wait for at least several months from now, in order for the “added” experience on SARS-CoV-2 and respiratory coinfections to be published and therefore analyzed. Unfortunately, this option is not feasible, given that our review has been already submitted. However, we thank the reviewer for this particular comment. Indeed, we are thinking of reviewing new knowledge and present it in a new article.           

Specific comments on the paper

Specific comment 1 

The English standard is not high and the manuscript would be improved by editing by someone with English as their first language. Some word choices for example Abstract first line replace "largely" with "highly or severely"; Abstract mid paragraph, replace "nowadays" with "present times".

Answer to specific comment 1

We thank the reviewer for this comment. Accordingly, we went throughout the text and correct the English standard. In particular, in Abstract we replaced “largely” with “highly” and “nowadays” with “present times”.   

Specific comment 2   

An example of 1. is the continual misspelling of "systematic" throughout the manuscript.  The authors have peppered the manuscript with the incorrect term "systemic" eg last paragraph of the introduction but it occurs in a number of other places.

Answer to specific comment 2

We corrected “systemic” to “systematic” throughout the text. Thank you!

Specific comment 3

Either in the introduction or the discussion there needs to be some discussion about the idea of "detection" versus "infection".  As a large proportion of cases of dual "infection" were asymptomatic this raises the question of whether these were truly infections causing a level of disease or merely detection of the pathogen without clinical consequence.  A study of reports which give details of the relative Ct (cycle thresholds) might give some insight into this fact.  For example if the Ct's were low (ie high viral load) with COVID-19 and high Ct's for influenza (ie low viral load) then one might attribute the clinical symptoms to be mainly due to COVID-19 at the time of infection. Of course the timelines for both "infections" might influence these Ct values.

Answer to specific comment 3

We highly agree with this comment. For this reason, we added a comment in “Implications for upcoming seasons” as well as in “Limitations” section. 

Specific comment 4   

Introduction 3rd line. Add "s" to "death" and this comment should be qualified as most hospitalizations and deaths are in the elderly especially in the over 80 year olds. Introduction last line on the first page, influenza "can effect up to 20% of the population".....   However it rarely does and is usually much lower in the 2-5% range!!

Answer to specific comment 4

Introduction, 3rd line: As recommended, we added “s” to “death” and we clarified that hospitalizations and deaths concern especially individuals >80 years old. Regarding influenza, we checked again our reference (WHO), and we can confirm that the above statement is correct (infecting up to 20%). However, we clarified this, by adding “depending on which viruses are circulating”.     

Specific comment 5    

Towards the end of the introduction the authors talk about vaccination programs.  It might also be worth mentioning that in the near future vaccines for the elderly and newborns (via maternal vaccination) will be available to add to the list of influenza and COVID-19 vaccines with this added protection but also potentially added logistical issues with the roll out of another respiratory virus vaccine unless combination vaccines are available.

Answer to specific comment 5

We thank the reviewer for pointing out this issue. Accordingly, we added a comment on the upcoming vaccines. We also added a comment that logistical issues should be addressed (End of Introduction section).   

Specific comment 6   

The methods the authors have used are a major issue.  They have included many reports with just a single case or just a few cases.  I think these should be deleted from the study as they add very little to the overall conclusions and may sway some figures eg the figure on page 15 which states that the mean percentage of co-infected patients with COVID-19 and influenza was "as low as 0.04% to 100%" a non-sensical conclusion! Eliminating these very low cases will have the added benefit of reducing the massive tables. One suggestion might be to only include studies with say >10 co-infections unless the total subjects analysed were say 50 or 100 subjects with COVID-19.

Answer to specific comment 6

We thank the reviewer for this comment. We recalculated the mean percentage of co-infected patients with influenza excluding isolated cases – case reports. We would like to ask the permission of the editor to keep all cases in the table (including isolated case reports). However, in order to minimize the length of the manuscript, and in accordance with reviewer’ 2 suggestions, we resubmit both tables as Supplementary Files 1 and 2.      

Specific comment 7

The data in the tables 1 and 2 should also be minimized in the main paper and the full data included in supplementary tables.

Answer to specific comment 7

We thank the reviewer for this suggestion. In response to comment No 6 and 7 of Reviewer 1, and to Reviewer 2 as well, we decided to keep the information in these tables and resubmit them as Supplementary Files 1 and 2. This way, the reader will still have access to the detailed information, while the manuscript will be reduced significantly.    

Specific comment 8   

In the discussion/conclusions apart from recommending vaccinations for COVID-19 and influenza and more multiplex testing, I think the authors should also be recommending the increased use of masks (especially N95/P2 masks if possible) during periods of increased activity of respiratory viruses or during influenza/RSV/COVID-19 "seasons" by those who are vulnerable (elderly/immunocompromised especially) and especially by all medical staff and frontline hospital staff. 

Answer to specific comment 8

We fully agree with this comment. We added relevant comments to recommend the use of masks during periods of increased activity of respiratory viruses by vulnerable individuals and healthcare personnel (“Implications for upcoming seasons” section and “Conclusions section”). 

Reviewer 2 Report

the article: COVID-19 and respiratory virus co-infections: a systemic review of the literature, is well presented and explains what was expected overall it contain a lot f information and interesting interlinks, future prospects and expectations but I do have some major and minor comments.

first of all, I would request for a chart for methodology section, where authors have searched for various topics.

Table 1 needs to be think over again and shrink.

a table showing all relavent list of diagnostic tests and their purpose will make understanding even better.

altogether the article is too long to be accepted or this sort of information. I strongly recommend it to be precise and efficient in presentation.

Minor errors

1. errors in flow diagram 1,

2. punctuation errors,

uniformity in table 1 is missing, some words are capital, some are not.

Author Response

Answer to Reviewer 2

Comments and Suggestions for Authors

Comment 1

the article: COVID-19 and respiratory virus co-infections: a systemic review of the literature, is well presented and explains what was expected overall it contain a lot f information and interesting interlinks, future prospects and expectations but I do have some major and minor comments.

first of all, I would request for a chart for methodology section, where authors have searched for various topics.

Answer to comment 1 

We thank the reviewer for the valuable efforts to improve our manuscript. Given that the PRISMA guidelines were used, the flow of literature search is described in details in Methods and presented in Figure 1.   

 Comment 2

Table 1 needs to be think over again and shrink.

Answer to Comment 2

We agree with both reviewers comments about Tables 1 and 2 being too long. Therefore, we decided to resubmit them as Supplementary Files 1 and 2. This way, the manuscript is reduced significantly, while the details are still available for the reader.        

Comment 3

a table showing all relavent list of diagnostic tests and their purpose will make understanding even better. altogether the article is too long to be accepted or this sort of information. I strongly recommend it to be precise and efficient in presentation.

Answer to Comment 3

We thank the reviewer particularly for this comment. Nonetheless, the list of all diagnostic tests is out of scope of this review and it will add too much text to our review. However, we chose to review only laboratory-confirmed cases. In order to reduce the length of the manuscript, we decided to resubmit both Tables 1 and 2 as Supplementary Files 1 and 2.          

Minor errors

Minor error 1

errors in flow diagram 1,

Answer to minor error 1 

Could the reviewer be more precise and indicate the error in flow diagnram? Thank you so much in advance.

Minor error 2

punctuation errors,

Answer to minor error 2

Punctuation errors were corrected throughout the text. Thank you!

 Minor error 3

uniformity in table 1 is missing, some words are capital, some are not.

Answer to minor error 3

We thank the reviewer for pointing this issue. We corrected Table 1 accordingly (resubmitted as Suplementary File 1).   

Comments of the Academic Editor

Dear authors,

Thank you again for your contributions. Please note that our academic editor has comments for you:

"In particular, it is recommended that the review is updated to encompass all of 2022, not just up until April 2022 - which is now almost a year ago. Additional data will surely have emerged in that time frame."

Response of the authors

We thank the editor for the thoughtful comment, which is in line with the comments of reviewer 1. We looked for papers on co-infection, that came out after April 12, 2022, during the Omicron wave and we found around 200 relevant papers. The huge number of new studies makes really hard to incorporate them in the current study, but we could use them for a follow-up paper, describing the particularities of the Omicron wave in terms of co-infections.

Yet, this is indeed a limitation of the paper, that we discuss extensively in the “Limitations” paragraph of our Discussion. We added the following phrase:

“We reviewed the literature up until April 12, 2022, when the Omicron wave was still in the beginning. This study was not able to detect any particularities concerning co-infections of patients infected by the Omicron variants and sub-variants”.

Round 2

Reviewer 1 Report

The authors have made a number of corrections to the ms according to my suggestions.  There are however a couple of matters that need addressing, especially the data analysis and supp tables.

Comment 1.

Line 53, With regard to Comment 4 about influenza attack rates. The reference 6 the authors have used is taken from a summary page on a WHO website that is NOT referenced to a peer reviewed journal and is therefore NOT acceptable.  I would suggest that the study by Jayasundara, K., Soobiah, C., Thommes, E. et al. Natural attack rate of influenza in unvaccinated children and adults: a meta-regression analysis. BMC Infect Dis 14, 670 (2014). https://doi.org/10.1186/s12879-014-0670-5, gives a much more realistic figures ie: "The attack rates (95% CI (Confidence Interval)) in adults for all influenza, type A and type B were 3.50% (2.30%, 4.60%), 2.32% (1.47%, 3.17%) and 0.59% (0.28%, 0.91%) respectively. For children, they were 15.20% (11.40%, 18.90%), 12.27% (8.56%, 15.97%) and 5.50% (3.49%, 7.51%) respectively" these are much more realistic estimates.

I would suggest using the lower-upper ranges derived in this article rather than using the 20% figure that the authors are insisting on, which I consider is misleading.

Line 83 Please modify to read ......against Respiratory syncytial virus.....

Comment 2.

I am having trouble now working out what the authors have done with the data tables and calculations of co-infections etc from the articles.  In figure 1 they show that some 129 articles were included in the study.  Supplementary table 1 which is COVID+influenza seems to contain some 45 studies 13 of which were single case studies.  Supp table 2 which is COVID+non influenza cases contains 37 studies with 6 single case studies. The total number of articles would then appear to be around 82 what happened to the other 47??

Then in the revised ms line 156 the authors state "overall out of 21 published studies with available data......" How did the number get to 21?  Presumably studies with low numbers were excluded but I cant see any mention of this in the text??

A full explanation is required of which studies were included/excluded for the calculations and why. Eg studies were only included in the calculations if there were a minimum of X subjects in the study.

Also there has been no recalculation of the COVID-19 with non-influenza viruses to exclude single cases/very small studies.  From supp table 2 there were 6 studies with just a single case. Why was this not done? These studies should be included/excluded using the same criteria as was applied to the COVID-19+influenza studies.

Can I suggest marking on each of these studies in supp tables 1 and 2 if they were included or not in the calculations. This could simply done by adding "included" or "excluded" under the study/country heading.

Author Response

The authors have made a number of corrections to the ms according to my suggestions.  There are however a couple of matters that need addressing, especially the data analysis and supp tables.

Comment 1

Line 53, With regard to Comment 4 about influenza attack rates. The reference 6 the authors have used is taken from a summary page on a WHO website that is NOT referenced to a peer reviewed journal and is therefore NOT acceptable.  I would suggest that the study by Jayasundara, K., Soobiah, C., Thommes, E. et al. Natural attack rate of influenza in unvaccinated children and adults: a meta-regression analysis. BMC Infect Dis 14, 670 (2014). https://doi.org/10.1186/s12879-014-0670-5, gives a much more realistic figures ie: "The attack rates (95% CI (Confidence Interval)) in adults for all influenza, type A and type B were 3.50% (2.30%, 4.60%), 2.32% (1.47%, 3.17%) and 0.59% (0.28%, 0.91%) respectively. For children, they were 15.20% (11.40%, 18.90%), 12.27% (8.56%, 15.97%) and 5.50% (3.49%, 7.51%) respectively" these are much more realistic estimates.

I would suggest using the lower-upper ranges derived in this article rather than using the 20% figure that the authors are insisting on, which I consider is misleading.

Line 83 Please modify to read ......against Respiratory syncytial virus.....

Answer to Comment 1

We thank the reviewer for the valuable comments. Accordingly, we replaced the text (in Introduction) and replaced the (old) reference [6] with the following reference: Jayasundara, K., Soobiah, C., Thommes, E. et al. Natural attack rate of influenza in unvaccinated children and adults: a meta-regression analysis. BMC Infect Dis 14, 670 (2014). We added the lower –upper ranges of attack rate, as requested by the reviewer. 

Line 83: We modified the sentence to read “… against RSV …”. Thank you for bringing this comment to our attention. 

Comment 2

  1. I am having trouble now working out what the authors have done with the data tables and calculations of co-infections etc from the articles.  In figure 1 they show that some 129 articles were included in the study.  Supplementary table 1 which is COVID+influenza seems to contain some 45 studies 13 of which were single case studies.  Supp table 2 which is COVID+non influenza cases contains 37 studies with 6 single case studies. The total number of articles would then appear to be around 82 what happened to the other 47??

Answer to point 1. We thank the reviewer for this particular comment. We included a total of 129 articles in our reference list. Among them there are 44 articles [9-15,25,27,34-68] presenting original data on SARS-CoV-2 and influenza co-infections (included in Supplementary Table 1) and 37 articles [18,22-24,54,59,66,67,75-103] presenting original data on respiratory virus co-infections other than influenza (presented in Supplementary Table 2). However, please note that four references [54,59,66,67] present data about influenza and other respiratory viruses SARS-CoV-2 infections, and therefore, are discussed in both sections (and Supplementary Tables 1and 2) . We also included additional articles (as mentioned in Methods), based on the relevance to our article. In particular, there are 8 articles presenting data on Animals’ Models [references 106-113]; 11 articles to present data about multiplex PCR and the importance of surveillance [118-128]; 20 articles in Introduction presenting generalized data about COVID-19 and respiratory viruses epidemiology [references 1-8,16,17,19-21,26,28-33]; 6 articles including one meta-analysis [References 69-74] to discuss issues of COVID-19 and influenza epidemiology and of bacterial co-infections (in the section of SARS-CoV-2 and influenza co-infections); 2 articles [references 104,105] concern a meta-analysis and data on bacterial and viral co-infections; 4 articles (references 114-117) to discuss issues of pathogenesis and severity (“Implications for upcoming seasons” section]; and 1 article (reference 129) was used in limitations section. Please note that there are some articles identified among the first selected 84 articles (e.g. one review and one on bacterial and respiratory viruses co-infections) that were included in the reference list but not in the original data about SARS-CoV-2 and respiratory viruses co-infections (Supplementary Tables 1 and 2).    

  1. Then in the revised ms line 156 the authors state "overall out of 21 published studies with available data......" How did the number get to 21?  Presumably studies with low numbers were excluded but I cant see any mention of this in the text??

Answer to point 2. As recommended in the first review round, we decided to include in the exportation of statistical measures (e.g. mean etc) only original studies presenting at the same time the denominator (=how many were tested and how many were found positive for influenza and COVID-19?). This way we ended with a total of 21 articles. This information was added in the text (Methods, page 3).  

  1. A full explanation is required of which studies were included/excluded for the calculations and why. Eg studies were only included in the calculations if there were a minimum of X subjects in the study.

Answer to point 3. Please read above our answer to comment 2. Thank you so much!!

  1. Also there has been no recalculation of the COVID-19 with non-influenza viruses to exclude single cases/very small studies.  From supp table 2 there were 6 studies with just a single case. Why was this not done? These studies should be included/excluded using the same criteria as was applied to the COVID-19+influenza studies.

Answer to point 4. We thank the reviewer for this comment. Please note that we did not exclude studies with 1 case (or case series) from Supplementary Table 1 and 2. However, we excluded case studies and case series from the calculation of mean and median prevalence of influenza and COVID-19 co-infections (please read our answer to comment 2). We would like to notice that no mean and median estimates were calculated for SARS-CoV-2 and other respiratory viruses co-infections (other than influenza) given that, in our opinion, there is no scientific rationale for doing so for a wide spectrum of viruses co-infections.    

  1. Can I suggest marking on each of these studies in supp tables 1 and 2 if they were included or not in the calculations. This could simply done by adding "included" or "excluded" under the study/country heading.

Answer to point 5. Given that a lot of articles are already presented in Supplementary Tables 1 and 2, we believe that if we add any additional information will end to a mess and not facilitate readers. However, in the main manuscript, all references appear when a mean, median or other calculations (or even qualitative estimations) are presented (e.g. CT findings). Please note that we highlighted these references in yellow color in order to facilitate the reviewer.